# Practical Time-Varying Formation Control for Nonlinear Multiagent Systems with Event-triggered Mechanism

Shuxing Xuan
*School of Automation Engineeringon*
*University of Electronic Science and Technology of China*
Chengdu, China
xsxlut@163.com

*Abstract*—This paper addresses the problem of practical time-varying formation control for nonlinear multi-agent systems with an event-triggered mechanism. In dynamic and complex multi-agent systems, maintaining a desired formation while adapting to nonlinearities and varying communication constraints presents significant challenges. We propose a novel control strategy that incorporates a time-varying formation criterion and utilizes an event-triggered mechanism to optimize communication efficiency.Our approach begins with the design of a practical time-varying formation control law tailored to handle the nonlinear dynamics inherent in multi-agent systems. The event-triggered mechanism is employed to reduce the frequency of control updates and communication overhead by activating the control updates only when necessary, based on predefined criteria. This mechanism ensures that the agents adjust their control actions in response to significant deviations from the desired formation, thus maintaining formation integrity while conserving communication resources.We derive stability conditions for the proposed control strategy and demonstrate its effectiveness through rigorous theoretical analysis and simulations. Theoretical results confirm that the system can achieve and maintain the desired time-varying formation despite the nonlinearities and dynamic communication constraints. Simulation experiments validate the practicality and robustness of the proposed method, showing that it effectively maintains the formation and reduces communication load.

*Index Terms*—Time-Varying formation, event-triggered, nonlinear

## I. Introduction

In the study of multi-agent systems, the problem of formation control has become a significant topic, especially in practical applications such as drone formations, autonomous vehicles, and robot fleets. Ensuring that agents can maintain a desired time-varying formation while addressing nonlinear dynamics and dynamic communication constraints is a challenging task.

Traditional formation control methods often assume that the system's dynamics are linear or that communication remains stable and continuous throughout the formation process. However, in practical scenarios, interactions between agents frequently exhibit nonlinear characteristics, and communication resources may be constrained or unstable. Therefore, designing

Identify applicable funding agency here. If none, delete this.

an effective control strategy to address these challenges while achieving time-varying formation goals is an urgent problem.

In recent years, event-triggered mechanisms have been proposed as an efficient communication strategy to optimize control performance in multi-agent systems. This mechanism significantly reduces communication overhead by triggering control updates only when necessary. However, integrating event-triggered mechanisms with time-varying formation control to address nonlinear dynamics in practical situations has not been thoroughly explored.

To address these issues, this paper proposes a novel control strategy that combines practical time-varying formation control with an event-triggered mechanism to handle nonlinear dynamics and communication constraints in multi-agent systems. First, we design a control law tailored for time-varying formations and use the event-triggered mechanism to optimize the frequency of control updates. Second, we validate the effectiveness and robustness of the proposed method through theoretical analysis and simulation experiments. The results demonstrate that the proposed method effectively maintains the desired formation while significantly reducing communication overhead.

The main contribution of this paper lies in providing a comprehensive approach that integrates time-varying formation control with event-triggered mechanisms, offering a new solution to practical control problems in multi-agent systems and laying a foundation for future research directions.

## II. Problem Formulation and Preliminaries

### A. Graph Theory

A signed digraph $\mathcal{G} = \{\mathcal{V}, \mathcal{E}, \mathcal{A}\}$ represents the communication topology of the multiagent systems, which includes $N$ followers and $M$ leaders. The $\mathcal{V} = \{v_1, \ldots, v_{N+M}\}$ denotes the node set, $\mathcal{E} = \{(v_i, v_j)|v_i, v_j \in \mathcal{V}\}$ represents the edge set, and $\mathcal{A} = [a_{ij}] \in \mathbb{R}^{(N+M) \times (N+M)}$ is the adjacency matrix for digraph $\mathcal{G}$. If $(v_i, v_j) \in \mathcal{E}$, it indicates that information can be transmitted from $j$ to $i$ with $a_{ij} = 1$; otherwise, $a_{ij} = 0$. Additionally, both competition and cooperation are considered in the communication topology. A positive value $(a_{ij} > 0)$ indicates cooperation, while a negative value

$(a_{ij} < 0)$ indicates competition. The adjacency matrix is denoted as $\mathcal{L} = \mathcal{D} - \mathcal{A}$, where $\mathcal{D} = \mathrm{diag}(d_1, \ldots, d_{N+M})$ with $d_i = \sum_{j=1}^{n} |a_{ij}|$. Then, define $\mathcal{B} = \mathrm{diag}\{|b_1|, |b_1|, \ldots, |b_N|\}$, where $b_i$ represents the weights between followers and leaders. If $i$th agent can receive information from a leader, then $b_i \neq 0$; otherwise, $b_i = 0$.

### B. Fuzzy Logic Systems

Fuzzy logic systems (FLSs) are frequently used to handle unmodeled dynamics in systems. As shown in [**?**], a FLSs can be described as

$$y = \frac{\sum_{l=1}^{m} \omega_l \left( \prod_{i=1}^{p} \mu_{V_i^l}(x_i) \right)}{\sum_{l=1}^{m} \left( \prod_{i=1}^{p} \mu_{V_i^l}(x_i) \right)} \qquad (1)$$

where $x = [x_1, \cdots, x_n]^\top$ is the input variable, $y$ is the output variable, and $\mu_{V_i^l}(x_i)$ $(i = 1, \cdots, p)$ is the membership function of $x_i$, $m$ is the number of rules.

The fuzzy basis function can be described as

$$\psi_l(\mathbf{x}) = \frac{\prod_{i=1}^{p} \mu_{V_i^l}(x_i)}{\sum_{l=1}^{m} \left( \prod_{i=1}^{p} \mu_{V_i^l}(x_i) \right)}. \qquad (2)$$

### C. Preliminaries

Consider nonstrict-feedback multiagent systems as follows:

$$\begin{cases} \dot{x}_{i,k} = x_{i,k+1} + f_{i,k}(\bar{x}_{i,k}) \\ \dot{x}_{i,n} = u_i + f_{i,n}(\bar{x}_{i,n}) \\ y_i = x_{i,1} \end{cases} \qquad (3)$$

where $k = 1, \ldots, n-1$, $\bar{x}_{i,m} = [x_{i,1}, \ldots, x_{i,m}]^\top \in \mathbb{R}^m$, $(m = 1, \ldots, n)$ is the state vector, $f_{i,m}(\bar{x}_{i,m})$ represents unknown smooth nonlinear function. $u_i \in \mathbb{R}$ and $y_i \in \mathbb{R}$ denotes the system state input and output, respectively.

## III. Adaptive Prescribed-Time Formation Tracking Controller Design

In this section, we investigate the problem of adaptive prescribed-time formation tracking, examining the dynamic trade-off between achieving faster stabilization and conserving communication resources for any given settling time.

### A. Event-Triggered Prescribed-Time Controller Design

The time-varying formation of the multiagent systems (3) is described by a bounded continuous differentiable function $\xi_i$, which represents the distance between $i$th agent and the leader.

According to Definition 2, given any setting time $T_p$, it is satisfied that

$$\lim_{t \to T_p} \|y_i - \xi_i - y_d\| \in \mathcal{M} \qquad (4)$$

the systems (3) achieves formation tracking. $\mathcal{M}$ represents the prescribed-time attractive set.

Define the following coordinate transformation:

$$\begin{aligned} z_{i,1} &= \sum_{j=1}^{N} |a_{i,j}| \left( y_i - \xi_i - \mathrm{sgn}(a_{i,j}) y_j + \xi_j \right) \\ &\quad + b_i (y_i - \mathrm{sgn}(b_i) y_d - \xi_i) \end{aligned} \qquad (5)$$

$$z_{i,q} = x_{i,q} - \eta_{i,q-1}$$

where $q = 2, \ldots, n$, $z_{i,1}$ is formation error, and $\eta_{i,q-1}$ represents the virtual controller to be designed later.

*Step 1:* From the systems (3) and coordinate transformation (5), the time derivative of $z_{i,1}$ is

$$\begin{aligned} \dot{z}_{i,1} &= \left( \sum_{j=1}^{N} |a_{i,j}| + b_i \right)(z_{i,2} + \eta_{i,1} + f_{i,1}(x_{i,1}) - \dot{\xi}_i) \\ &\quad - \sum_{j=1}^{N} |a_{i,j}| \left( \mathrm{sgn}(a_{i,j}) \dot{y}_j - \dot{\xi}_j \right) - b_i \mathrm{sgn}(b_i) \dot{y}_d \end{aligned} \qquad (6)$$

where $F_{i,1}(X_{i,1}) = f_{i,1}(x_{i,1}) - \frac{1}{\hbar_i}(\sum_{j=1}^{N} |a_{i,j}| (\mathrm{sgn}(a_{i,j}) \dot{y}_j - \dot{\xi}_j) - b_i \mathrm{sgn}(b_i) \dot{y}_d)$ with $X_{i,1} = [x_{i,1}, x_{j,1}, y_d, \dot{y}_d, \dot{\xi}_j]^\top$, and $\hbar_i = \sum_{j=1}^{N} |a_{i,j}| + b_i$. FLSs can be applied to approximate the unknown nonlinear dynamic $F_{i,1}(X_{i,1})$ as $F_{i,1}(X_{i,1}) = W_{i,1}^\top \psi_{i,1}(X_{i,1}) + \varepsilon_{i,1}(X_{i,1})$, it can be obtained

$$\dot{z}_{i,1} = \hbar_i(z_{i,2} - \dot{\xi}_i + \eta_{i,1} + W_{i,1}^\top \psi_{i,1}(X_{i,1}) + \varepsilon_{i,1}(X_{i,1})). \qquad (7)$$

Based on the coordinate transformation (5), the candidate Lyapunov function is selected as

$$V_{i,1} = \frac{1}{2} z_{i,1}^2 + \frac{1}{2} \tilde{\theta}_{i,1}^2 \qquad (8)$$

where $\tilde{\theta}_{i,1} = \theta_{i,1} - \hat{\theta}_{i,1}$ is the estimate error, $\hat{\theta}_i$ is the estimated value of the uncertain parameter $\theta_i$ with $\theta_{i,1} = \|W_{i,1}\|^2$.

According to the derivative of $z_{i,1}$ in (7) and the candidate Lyapunov function (8), one yileds

$$\begin{aligned} \dot{V}_{i,1} &= z_{i,1}(\hbar_{i,1}(X_{i,1}) + \varepsilon_{i,1}(X_{i,1}))) \\ &\quad + z_{i,1} \hbar_i(\eta_{i,1} - \dot{\xi}_i) - \tilde{\theta}_{i,1} \dot{\hat{\theta}}_{i,1} \end{aligned} \qquad (9)$$

By employing Young's inequality, it can be inferred

$$\begin{aligned} &z_{i,1} \hbar_i \psi_{i,1}(X_{i,1}) + \varepsilon_{i,1}(X_{i,1})) \leq \\ &\frac{1}{2} \epsilon_{i,1} z_{i,1}^2 \hbar_i \theta_{i,1} \|\psi_{i,1}(X_{i,1})\|^2 \\ &+ \frac{\hbar_i}{2 \epsilon_{i,1}} + \frac{1}{2} \hbar_i \bar{\varepsilon}_{i,1}^2 + \frac{1}{2} \hbar_i z_{i,1}^2 \end{aligned} \qquad (10)$$

where $\epsilon_{i,1} > 0$ is the design parameter, $|\varepsilon_{i,1}(X_{i,1})| \leq \bar{\varepsilon}_{i,1}$ with $\bar{\varepsilon}_{i,1} > 0$ is a constant.

The virtual controller and adaptive law are designed as

$$\begin{aligned} \eta_{i,1} &= \frac{1}{\hbar_i}(-\mu(t) k_{i,11} sig^{2\alpha-1}(z_{i,1}) - \frac{1}{2} \hbar_i^2 z_{i,1} \\ &\quad - \mu(t) k_{i,12} sig^{2\beta-1}(z_{i,1}) - k_{i,13} sig^{2\beta-1}(z_{i,1}) \\ &\quad - \frac{1}{2} \epsilon_{i,1} z_{i,1} \hbar \psi^2 + \dot{\xi}_i) \end{aligned} \qquad (11)$$

$$\begin{aligned} \dot{\hat{\theta}}_{i,1} &= \frac{1}{2} \epsilon_{i,1} z_{i,1}^2 \hbar \psi_{i,1}(X_{i,1})^2 - \mu(t) a_{\theta i,1} \hat{\theta}_{i,1} \\ &\quad - \mu(t) b_{\theta i,1} \hat{\theta}_{i,1}^{2\beta-1} - c_{\theta i,1} \hat{\theta}_{i,1}^{2\beta-1} \end{aligned} \qquad (12)$$

where $k_{i,11}, k_{i,12},$ and $k_{i,13} > 0$, $a_{\theta i,1}, b_{\theta i,1},$ and $c_{\theta i,1} > 0$ are design parameters.

Inserting (10), (11) and (12) into (9), it can be obtained

$$
\begin{aligned}
\dot{V}_{i,1} \leq &-\mu(t)z_{i,1}k_{i,11}sig^{2\alpha-1}(z_{i,1}) - \mu(t)z_{i,1}k_{i,12}sig^{2\beta-1}(z_{i,1}) \\
&- z_{i,1}k_{i,13}sig^{2\beta-1}(z_{i,1}) + \hbar_i z_{i,1}z_{i,2} + c_{\theta i,1}\tilde{\theta}_{i,1}\hat{\theta}_{i,1}^{2\beta-1} \\
&+ \mu(t)b_{\theta i,1}\tilde{\theta}_{i,1}\hat{\theta}_{i,1}^{2\beta-1} + \mu(t)a_{\theta i,1}\tilde{\theta}_{i,1}\hat{\theta}_{i,1} \\
&+ \frac{\hbar_i}{2\epsilon_{i,1}} + \frac{1}{2}\hbar_i\bar{\varepsilon}_{i,1}
\end{aligned}
\tag{13}
$$

*Step $n$:*Inductively, one can obtain that the derivative of $z_{i,n}$ satisfies

$$
\dot{z}_{i,n} = u_i + f_{i,n}(\bar{x}_{i,n}) - \dot{\eta}_{i,n-1} \tag{14}
$$

where the unknown nonlinear function $F_{i,n}(X_{i,n}) = f_{i,n}(\bar{x}_{i,n}) - \dot{\eta}_{i,n-1}$ is defined, and a FLSs $W^\top\psi(x)$ is utilized to approximate the function $F_{i,n}(X_{i,n})$, such that $F_{i,n}(X_{i,n}) = W_{i,n}^\top\psi_{i,n}(X_{i,n}) + \varepsilon_{i,n}(X_{i,n})$ with $X_{i,n} = [\bar{x}_{i,n}, \dot{\eta}_{i,n-1}]^\top$. $|\varepsilon_{i,n}(X_{i,n})| \leq \bar{\varepsilon}_{i,n}$ with $\bar{\varepsilon}_{i,n} > 0$ is a constant.

By selecting the candidate Lyapunov function as

$$
V_{i,n} = V_{i,n-1} + \frac{1}{2}z_{i,n}^2 + \frac{1}{2}\tilde{\theta}_{i,n}^2 \tag{15}
$$

where the parameter estimation error satisfies $\tilde{\theta}_{i,q} = \theta_{i,q} - \hat{\theta}_{i,q}$, and $\theta_{i,q} = \|W_{i,q}\|^2$.

From (**??**) and (**??**), $\dot{V}_{i,q}$ satisfies

$$
\begin{aligned}
\dot{V}_{i,q} = &\dot{V}_{i,q-1} + z_{i,q}(z_{i,q+1} + W_{i,q}^\top\psi_{i,q}(X_{i,q}) + \varepsilon_{i,q}(X_{i,q})) \\
&+ z_{i,q}\eta_{i,q} - \tilde{\theta}_{i,q}\dot{\hat{\theta}}_{i,q}
\end{aligned}
\tag{16}
$$

By applying Young's inequality, it can be deduced

$$
\begin{aligned}
z_{i,q}(W_{i,q}^\top\psi_{i,q}(X_{i,q}) + \varepsilon_{i,q}(X_{i,q})) \leq \\
\frac{1}{2}\epsilon_{i,q}z_{i,q}^2\theta_{i,q}\|\psi_{i,q}(X_{i,q})\|^2 + \frac{1}{2\epsilon_{i,q}} + \frac{1}{2}\bar{\varepsilon}_{i,q}^2 + \frac{1}{2}z_{i,q}^2
\end{aligned}
\tag{17}
$$

where $\epsilon_{i,q} > 0$ is a design parameter, $|\varepsilon_{i,q}(X_{i,q})| \leq \bar{\varepsilon}_{i,q}$ with $\bar{\varepsilon}_{i,q} > 0$ is a constant.

### B. Stability Analysis

The principal theory introduced in this paper are proven to be effective by the following theorem.

For multiagent systems (3), if Assumptions 1-3 are satisfied, for any initial state $\bar{x}_n(0)$, the proposed virtual controllers, the parameter update laws, and the event-triggered controller ensure that all signals are prescribed time stable, and the bipartite formation tracking error $z_{i,1}$ satisfies:

Converges to to a small neighborhood $\mathcal{M}_1$ within the preset time $T_p$. Meanwhile, the Zeno behavior is circumvented.

Note that the method proposed in this paper considers the impact of convergence time on the system. In practical system control, a very short convergence time requires a larger control input to achieve stable operation. Additionally, input saturation needs to be considered in practical control systems. For the event-triggered scheme (**??**), it can be seen that the dynamic threshold is composed of the prescribed-time evolution function and input saturation constraints. The

parameter $D_k$ is used to balance convergence speed and system communication resource optimization. If the user sets a very short convergence time, a shorter update interval is adopted to achieve faster stabilization. When a longer convergence time is set, a longer update interval is used to better reduce communication resource consumption.

## IV. CONCLUSION

In this paper, we have developed a novel control strategy for practical time-varying formation control in nonlinear multi-agent systems, incorporating an event-triggered mechanism to optimize communication efficiency. Our approach addresses the dual challenges of maintaining a desired time-varying formation and managing nonlinear dynamics while reducing communication overhead. Through rigorous theoretical analysis, we derived stability conditions that guarantee the effectiveness of the proposed control strategy. Simulation results further validate the practicality and robustness of the method, demonstrating that it not only successfully maintains the desired formation but also ensures the boundedness of all signals within the closed-loop system. The proposed method effectively reduces communication load while achieving the control objectives, making it suitable for real-world applications where communication resources are limited and system dynamics are complex. In summary, this paper contributes a significant advancement in the field of multi-agent systems by providing a robust and efficient solution for time-varying formation control under nonlinear dynamics. Future work may explore extending this approach to more complex scenarios, such as systems with time delays or varying communication topologies, and further optimizing the event-triggered mechanism to enhance overall system performance.

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
