# OpenReview forum: "Practical Time-Varying Formation Control for Nonlinear Multiagent Systems with Event-triggered Mechanism"
_IEEE.org/ICIST/2024/Conference — IEEE ICIST 2024 Conference Submission_

### Official Review · Reviewer_SC5D · 2024-08-20
**Manuscript Rejection**

**Rating:** 3
**Confidence:** 4

**Review:**

The paper does not sufficiently differentiate its proposed strategy from existing methods in the literature. The control strategy and event-triggered mechanism are concepts that have been explored extensively. Without a robust comparison to state-of-the-art methods, the claimed novelty and advantages of the approach are not convincingly demonstrated.

While stability conditions are derived, the theoretical analysis lacks depth in explaining how the proposed mechanism handles specific challenges posed by the time-varying constraints. More rigorous theoretical proof or a detailed exploration of edge cases would strengthen the contribution.

---

### Official Review · Reviewer_zb3G · 2024-08-21
**Manuscript Rejection**

**Rating:** 3
**Confidence:** 4

**Review:**

The author proposes a new control strategy for the nonlinear multi-agent systems with an event-triggered mechanism in this paper. However, the simulation section does not provide, and the advantages of the new control method lack depth exploration.
The author need to emphasize and describe the advantages of the proposed control scheme compared to existing achievements.

---

### Official Review · Reviewer_D6hM · 2024-08-21
**Recommended rejection**

**Rating:** 3
**Confidence:** 3

**Review:**

The manuscript lacks sufficient novelty and does not provide a comprehensive comparison with existing formation control strategies, limiting its overall contribution to the field.  As such, I do not think this paper is suitable for ICIST 2024.

---

### Decision · Program_Chairs · 2024-09-08

Reject